# Estimating Entropy of Distributions in Constant Space

**Jayadev Acharya**
Cornell University
acharya@cornell.edu

**Sourbh Bhadane**
Cornell University
snb62@cornell.edu

**Piotr Indyk**
Massachusetts Institute of Technology
indyk@mit.edu

**Ziteng Sun**
Cornell University
zs335@cornell.edu

## Abstract

We consider the task of estimating the entropy of $k$-ary distributions from samples in the streaming model, where space is limited. Our main contribution is an algorithm that requires $O\left(\frac{k \log(1/\varepsilon)^2}{\varepsilon^3}\right)$ samples and a constant $O(1)$ memory words of space and outputs a $\pm\varepsilon$ estimate of $H(p)$. Without space limitations, the sample complexity has been established as $S(k,\varepsilon) = \Theta\left(\frac{k}{\varepsilon \log k} + \frac{\log^2 k}{\varepsilon^2}\right)$, which is sub-linear in the domain size $k$, and the current algorithms that achieve optimal sample complexity also require nearly-linear space in $k$.

Our algorithm partitions $[0, 1]$ into intervals and estimates the entropy contribution of probability values in each interval. The intervals are designed to trade off the bias and variance of these estimates.

## 1 Introduction

**Streaming Algorithms.** Algorithms that require a limited memory/space/storage[1] have garnered great interest over the last two decades, and are popularly known as *streaming algorithms*. Initially studied by [1, 2], this setting became mainstream with the seminal work of [3]. Streaming algorithms are particularly useful in handling massive datasets that are impossible to be stored in the memory of the system. It is also applicable in networks where data is naturally generated sequentially and the data rates are much higher than the capabilities of storing them, e.g., on a router.

The literature on streaming algorithms is large, and many problems have been studied in this model. With roots in computer science, a large fraction of this literature considers the worst case model, where it is assumed that the input $X^n := X_1, \ldots, X_n$ is an arbitrary sequence over some domain of size $k$ (e.g., $[k] := \{1, \ldots, k\}$). The set-up is as follows:

*Given a system with limited memory that can make a few (usually just one) passes over $X^n$, the objective is to estimate some $f(X^n)$ of the underlying dataset. The primary objective is solving the task with as little memory as possible, which is called the* space complexity.

Some of the research closest to our task is the estimation of frequency moments of the data stream [3, 4, 5], the Shannon and Rényi entropy of the empirical distribution of the data stream [6, 7, 8, 9, 10], the heavy hitters [11, 12, 13, 14], and distinct elements [15, 16]. There has also been work on random order streams, where one still considers a worst case data stream $X^n$, but feeds a random permutation $X_{\sigma(1)}, \ldots, X_{\sigma(n)}$ of $X^n$ as input to the algorithm [10, 17, 18].

**Statistical Estimation.** At the same time, there has been great progress in the classical fields of statistical learning and distribution property estimation. The typical set-up is as follows:

*Given independent samples $X^n$ from an unknown distribution $p$, the objective is to estimate a property $f(p)$ using the fewest samples, called the* sample complexity.

Distribution property estimation literature most related to our work include entropy estimation [19, 20, 21, 22, 23, 24, 25], support size estimation [21, 23, 26], Rényi entropy estimation [27, 28, 29], support coverage estimation [30, 31], and divergence estimation [32, 33]. In these tasks, the optimal sample complexity is *sub-linear* in $k$, the domain size of the distribution.

**Streaming Algorithms for Statistical Estimation.** While space complexity of streaming algorithms, and sample complexity of statistical estimation have both received great attention, the problem of statistical estimation under memory constraints has received relatively little attention. Interestingly, almost half a century ago, Cover and Hellman [34, 35] studied this setting for hypothesis testing with finite memory, and [36] had studied estimating the bias of a coin using a finite state machine. However, until recently, there are few works on learning with memory constraints. There has been a recent interest in space-sample trade-offs in statistical estimation [37, 38, 39, 40, 41, 42, 43, 44]. Within these, [40] is the closest to our paper. They consider estimating the integer moments of distributions, which is equivalent to estimating Rényi entropy of integer orders under memory constraints. They present natural algorithms for the problem, and perhaps more interestingly, prove non-trivial lower bounds on the space complexity of this task. More recently, [45] obtained memory sample trade-offs for testing discrete distributions.

We initiate the study of distribution entropy estimation with space limitations, with the goal of understanding the space-sample trade-offs.

## 1.1 Problem Formulation

Let $\Delta_k$ be the class of all $k$-ary discrete distributions over the set $\mathcal{X} = [k] := \{0, 1, \ldots, k-1\}$. The Shannon entropy of $p \in \Delta_k$ is $H(p) := -\sum_{x \in [k]} p(x) \log(p(x))$. Entropy is a fundamental measure of randomness and a central quantity in information theory and communications. Entropy estimation is a key primitive in various machine learning applications for feature selection.

Given independent samples $X^n := X_1, \ldots, X_n$ from an unknown $p \in \Delta_k$, an entropy estimator is a possibly randomized mapping $\widehat{H} : [k]^n \to \mathbb{R}$. Given $\varepsilon > 0, \delta > 0$, $\widehat{H}$ is an $(\varepsilon, \delta)$ estimator if

$$\forall p \in \Delta_k, \Pr_{X^n \sim p^{\otimes n}} \left( |\widehat{H}(X^n) - H(p)| > \varepsilon \right) < \delta, \tag{1}$$

where $p^{\otimes n}$ denotes the joint distribution of $n$ independent samples from $p$.

**Sample Complexity.** The *sample complexity* $S(H, k, \varepsilon, \delta)$ is the smallest $n$ for which an estimator satisfying (1) exists. Throughout this paper, we assume a constant error probability, say $\delta = 1/3$,[2] and exclusively study entropy estimation. We therefore denote $S(H, k, \varepsilon, 1/3)$ by $S(k, \varepsilon)$.

**Memory Model and Space Complexity.** The basic unit of our storage model is a *word*, which consists of $\log k + \log(1/\varepsilon)$ bits. This choice of storage model is motivated by the fact that at least $\log(1/\varepsilon)$ bits are needed for a precision of $\pm\varepsilon$, and $\log k$ bits are needed to store a symbol in $[k]$. The *space complexity* of an algorithm is the smallest space (in words) required for its implementation.

## 1.2 Prior Work

**Distribution Entropy Estimation.** Entropy estimation from samples has a long history [19, 46, 47]. The most popular method is empirical plug-in estimation that outputs the entropy of the empirical distribution of the samples. Its sample complexity [47, 20] is

$$S^e(k, \varepsilon) = \Theta \left( k/\varepsilon + (\log^2 k)/\varepsilon^2 \right). \tag{2}$$

Paninski [48] showed that there exists an estimator with sub-linear sample complexity in $k$. A recent line of work [21, 23, 22] has characterized the optimal sample complexity as

$$S(k, \varepsilon) = \Theta \left( k/(\varepsilon \log k) + \log^2 k/\varepsilon^2 \right). \tag{3}$$

Note that the optimal sample complexity is sub-linear in $k$, and that of empirical estimator is linear.

**Estimating Entropy of Streams.** There is significant work on estimating entropy of the stream with limited memory. Here, no distributional assumptions on the input stream $X^n$, and the goal is to estimate $H(X^n)$, the entropy of the empirical distribution of $X^n$. [6, 49, 10, 9, 8] consider multiplicative entropy estimation. These algorithms can be modified to additive entropy estimation by noting that $(1 \pm \varepsilon / \log n)$ multiplicative estimation is equivalent to a $\pm \varepsilon$ additive estimation. With this, [8, 10] give an algorithm requiring $O(\frac{\log^3 n}{\varepsilon^2})$ words of space for $\pm \varepsilon$ estimate of $H(X^n)$. [9] proposes an algorithm using $O(\frac{\log^2 n \cdot \log \log n}{\varepsilon^2})$ words of space. A space lower bound of $\Omega(1/\varepsilon^2)$ was proved in [8] for the worst-case setting.

Another widely used notion of entropy is Rényi entropy [50]. The Rényi entropy of $p$ of order $\alpha > 0$ is $H_\alpha(p) := \log(\sum_x p(x)^\alpha)/(1 - \alpha)$. [51, 52, 27] show that the sample complexity of estimating $H_\alpha(p)$ is $\Theta(k^{1-1/\alpha}/\varepsilon^2)$ for $\alpha \in \mathbb{N}$. [40] studies the problem of estimating the collision probability, which can be seen as estimating $H_\alpha(p)$ for $\alpha = 2$, under memory constraints. They propose an algorithm with sample complexity $n$ and the memory $M$ satisfying $n \cdot M \geq \Omega(k)$, when $n$ is at least $O(k^{1-1/\alpha})$. They also provide some (non-tight) lower bounds on the memory requirements.

### 1.3 Our Results and Techniques

We consider the problem of estimating $H(p)$ from samples $X^n \sim p$, with as little space as possible. Our motivating question is: *What is the space-sample trade-off of entropy estimation over $\Delta_k$?*

The optimal sample complexity is given in (3). However, straight-forward implementations of sample-optimal schemes in [21, 23, 22] require nearly linear space complexity in $S(k, \varepsilon)$, which is nearly linear (in $k$) words space. At the same time, when the number of samples is more than $S^e(k, \varepsilon)$, given in (2), the empirical entropy of $X^n$ is within $\pm \varepsilon$ of $H(p)$. We can therefore use results from streaming literature to estimate the empirical entropy of a data-stream with $n = S^e(k, \varepsilon)$ samples to within $\pm \varepsilon$, and in doing so, obtain a $\pm 2\varepsilon$ estimate of $H(p)$. In particular, the algorithm of [9] requires $S^e(k, \varepsilon)$ samples, and with $O(\frac{\log^2(S^e(k,\varepsilon)) \log \log(S^e(k,\varepsilon))}{\varepsilon^2})$ words of space, estimates $H(p)$ to $\pm \varepsilon$. Note that $S^e(k, \varepsilon)$ is linear with respect to $k$.

Our work requires constant words of space while maintaining linear sample complexity in $k$.

**Theorem 1.** *There is an algorithm that requires* $O\left(\frac{k(\log(1/\varepsilon))^2}{\varepsilon^3}\right)$ *samples and 20 words of space and estimates $H(p)$ to $\pm \varepsilon$.*

The results and the state of the art are given in Table 1. A few remarks are in order.

**Remark.** (1). Our algorithm can bypass the lower bound of $\Omega(1/\varepsilon^2)$ for entropy estimation of data-streams since $X^n$ is generated by a distribution and not the worst case data stream. (2). Consider the case when $\varepsilon$ is a constant, say $\varepsilon = 1$. Then, the optimal sample complexity is $\Theta(\frac{k}{\log k})$ (from (3)) and all known implementations of these algorithms requires $\tilde{\Theta}(k)$ space. Streaming literature provides an algorithm with $O(k)$ samples and $\tilde{O}((\log k)^2)$ memory words. We provide an algorithm with $O(k)$ samples, and 20 memory words. Compared to the sample-optimal algorithms, we have a $\log k$ blow-up in the sample complexity, but an exponential reduction in space.

Table 1: Sample and space complexity for estimating $H(p)$.

| Algorithm | Samples | Space (in words) |
|---|---|---|
| Sample-Optimal [21],[23, 22] | $\Theta\left(\frac{k}{\varepsilon \log k} + \frac{\log^2 k}{\varepsilon^2}\right)$ | $O\left(\frac{k}{\varepsilon \log k} + \frac{\log^2 k}{\varepsilon^2}\right)$ |
| Streaming [8, 9] | $O\left(\frac{k}{\varepsilon} + \frac{\log^2 k}{\varepsilon^2}\right)$ | $O\left(\log^2(\frac{k}{\varepsilon}) \log \log(\frac{k}{\varepsilon})/\varepsilon^2\right)$ |
| Algorithm 6 | $O\left(\frac{k(\log(1/\varepsilon))^2}{\varepsilon^3}\right)$ | 20 |

We now describe the high level approach and techniques. We can write $H(p)$ as

$$H(p) = -\sum_x p(x) \log p(x) = \mathbb{E}_{X \sim p}\left[-\log p(X)\right]. \tag{4}$$

**A Simple Method.** We build layers of sophistication to a simple approach: In each iteration,

1. Obtain a sample $X \sim p$.

2. Using constant memory, over the next $N$ samples, estimate $\log(1/p(X))$.

From (4), for large enough $N$, we can obtain a *good estimate* $\hat{p}(X)$ of $p(X)$, and $\log(1/\hat{p}(X))$ will be an almost unbiased estimate of the entropy. We can then maintain a running average of $\log(1/\hat{p}(X))$ over $R$ iterations, where $R$ is large enough for the empirical mean of $\log(1/\hat{p}(X))$ to concentrate. The total sample requirement is $NR$. This approach is described in Algorithm 1 (in Section 2). Theorem 4 states that it requires $O(1)$ memory words and the sample complexity is super-linear.

**Intervals for Better Sample Complexity.** To improve the sample complexity, we partition $[0, 1]$ into $T$ disjoint intervals (Algorithm 1 corresponds to $T = 1$). In Lemma 7 we express $H(p)$ as a sum of entropy-like expressions defined over probability values in these $T$ intervals. We will then estimate each of the terms separately with the approach stated above. We will show that the sample complexity as a function of $k$ drops down roughly as $k(\log^{(T)} k)^2$, where $\log^{(T)}$ is the $T$th iterated logarithm, while the space complexity is still constant. In the simple algorithm described above, we need

1. $N$, the number of samples for each iteration, to be large enough for good estimates of $p(X)$.

2. $R$, the number of iterations, to be large enough for concentration.

Note that when $p(X)$ is large, fewer samples are needed to estimate $p(X)$ (small $N$), and for small $p(X)$ more samples are needed. However, if the intervals are chosen such that small probabilities are also contained in small intervals, the number of iterations $R$ needed for these intervals can be made small (the range of random variables in Hoeffding's inequality is smaller). Succinctly,

*Fewer samples are needed to estimate the large probabilities, and fewer iterations are needed for convergence of estimates for small probabilities by choosing the intervals carefully.*

**Some Useful Tools.** We now state two concentration inequalities that we use throughout this paper.

**Lemma 2.** *(Hoeffding's Inequality)* [53] *Let $X_1, \ldots, X_m \in [a_i, b_i]$ be independent random variables. Let $X = (X_1 + \ldots + X_m)/m$, then $\Pr\left(|X - \mathbb{E}[X]| \geq t\right) \leq 2 \exp\left(\frac{-2(mt)^2}{\sum_i (b_i - a_i)^2}\right)$.*

In some algorithms we consider, $m$ itself is a random variable. In those cases, we will use the following variant of Hoeffding's inequality, which is proved in Section B.

**Lemma 3.** *(Random Hoeffding's Inequality) Let $M \sim Bin(m, p)$. Let $X_1, \ldots, X_m$ be independent random variables such that $X_i \in [a, b]$. Let $X = (\sum_{i=1}^{M} X_i)/M$. Then, for any $0 < p \leq 1$*

$$\Pr\left(|X - \mathbb{E}[X]| \geq t/p\right) \leq 3 \exp\left(-mt^2/(8p(b-a)^2)\right). \tag{5}$$

**Outline.** In Section 2 we describe the simple approach and its performance in Theorem 4. In Section 3.1, Algorithm 5 we show how the sample complexity can be reduced from $k \log^2 k$ in Theorem 4 to $k(\log\log k)^2$ in Theorem 8 by choosing two intervals ($T = 2$). The intervals are chosen such that the number of iterations $R$ for the small interval is $\text{poly}(\log\log k)$ in Algorithm 5 compared to $\text{poly}(\log k)$ in Algorithm 1. The algorithm for general $T$ is described in Section 3.2, and the performance of our main algorithm is given in Theorem 1.

## 2 A Building Block: Simple Algorithm with Constant Space

We propose a simple method (Algorithm 1) with the following guarantee.

**Theorem 4.** *Let $\varepsilon > 0$. Algorithm 1 takes $O\left(\frac{k \log^2(k/\varepsilon)}{\varepsilon^3}\right)$ samples from $p \in \Delta_k$, uses at most 20 words of memory, and outputs $\bar{H}$, such that with probability at least $2/3$, $\left|\bar{H} - H(p)\right| < \varepsilon$.*

Based on (4), each iteration of Algorithm 1 obtains a sample $X$ from $p$ and estimates $\log(1/p(X))$. To avoid assigning zero probability value to $p(X)$, we do add-1 smoothing to our empirical estimate of $p(X)$. The bias in our estimator can be controlled by the choice of $N$.

**Performance Guarantee.** Algorithm 1 only maintains a running sum at the end of each iteration. We reserve two words for $N$, $R$ and $S$. We reserve one word to store $x$ and two words to keep track

---

**Algorithm 1** Entropy estimation with constant space: Simple Algorithm

---

**Require:** Accuracy parameter $\varepsilon > 0$, a data stream $X_1, X_2, \ldots \sim p$
 1: Set
$$R \leftarrow 4\log^2(1 + 2k/\varepsilon)/\varepsilon^2, \qquad N \leftarrow 2k/\varepsilon, \qquad S \leftarrow 0$$
 2: **for** $t = 1, \ldots, R$ **do**
 3:     Let $x \leftarrow$ the next element in the data stream
 4:     $N_x \leftarrow$ # appearances of $x$ in the next $N$ symbols
 5:     $\hat{H}_t = \log\left(N/(N_x + 1)\right)$
 6:     $S = S + \hat{H}_t$
 7: $\bar{H} = S/R$

---

of $N_x$ in each iteration. We reserve three words for the counters. Thus the algorithm uses less than 20 words of space.

To bound the accuracy, note that $\bar{H}$ is the mean of $R$ i.i.d. random variables $\hat{H}_1, \ldots, \hat{H}_R$. We bound the bias and prove concentration of $\bar{H}$ using Lemma 2.

*Bias Bound.* Larger values of $N$ provides a better estimate of $p(X)$, and therefore a smaller bias in estimation. This is captured in the next lemma, which is proved in Section C.

**Lemma 5.** *(**Bias Bound**)* $\left|\mathbb{E}\left[\bar{H}\right] - H\left(p\right)\right| \leq \frac{k}{N}$.

*Concentration.* Since $\forall t, \hat{H}_t \in [\log(N/(N+1)), \log N]$, we show in the next lemma that with large enough $R$, $\bar{H}$ concentrates. This is proved in Section C.

**Lemma 6.** *(**Concentration**)* *For any* $\mu > 0$, $\Pr\left(|\bar{H} - \mathbb{E}\left[\bar{H}\right]| \geq \mu\right) \leq 2\exp\left(-\frac{2R\mu^2}{\log^2(N+1)}\right)$.

The choice of $N$ implies that $\left|\mathbb{E}\left[\bar{H}\right] - H\left(p\right)\right| \leq \varepsilon/2$, and by choosing $\mu = \varepsilon/2$, and $R = 4\log^2(1 + 2k/\varepsilon)/\varepsilon^2$ implies that $\bar{H}$ is within $H(p) \pm \varepsilon$ with probability at least $2/3$. The total sample complexity of Algorithm 1 is $(N + 1)R = O\left(k\log^2\left(k/\varepsilon\right)/\varepsilon^3\right)$.

## 3 Interval-based Algorithms

In the previous section, the simple algorithm treats each symbol equally and uses the same $N$ and $R$. To reduce the sample complexity, we express $H(p)$ as an expectation of various conditional expectations depending on the symbol probability values. For larger probability values we use smaller $N$ and for small probabilities we use smaller $R$. We then estimate the terms separately to obtain the final estimate.

**Entropy as a Weighted Sum of Conditional Expectations.** Let $T \in \mathbb{N}$ (decided later), and $0 = a_0 < a_1 < \ldots < a_T = 1$. Let $\mathcal{I} := \{I_1, I_2, ..., I_T\}$, where $I_j = [a_{T-j}, a_{T-j+1})$ be a partition of $[0, 1]$ into $T$ intervals.

Consider a randomized algorithm $\mathcal{A} : [k] \rightarrow \{I_1, \ldots, I_T\}$ that takes as input $x \in [k]$, and outputs an interval in $\mathcal{I}$. Let $p_{\mathcal{A}}\left(I_j|x\right) = \Pr\left(\mathcal{A}(x) = I_j\right)$. For a distribution $p \in \Delta_k$, let

$$p_{\mathcal{A}}(I_j) := \sum_{x \in [k]} p(x) \cdot p_{\mathcal{A}}\left(I_j|x\right), \qquad p_{\mathcal{A}}\left(x|I_j\right) := \frac{p(x) \cdot p_{\mathcal{A}}\left(I_j|x\right)}{p_{\mathcal{A}}\left(I_j\right)}. \tag{6}$$

Then $p_{\mathcal{A}}(I_j)$ is the probability that $\mathcal{A}(X) = I_j$, when $X \sim p$. $p_{\mathcal{A}}\left(x|I_j\right)$ is the conditional distribution over $[k]$ given $\mathcal{A}(X) = I_j$. Then we have the following lemma:

**Lemma 7.** *Let* $H_j := \mathbb{E}_{X \sim p_{\mathcal{A}}(x|I_j)}\left[-\log p\left(X\right)\right]$ *then,* $H\left(p\right) = \sum_{j=1}^{T} p_{\mathcal{A}}\left(I_j\right) H_j$.

*Proof.*

$$H\left(p\right) = \sum_x p(x)\left(\sum_j p_{\mathcal{A}}\left(I_j|x\right)\right)\log\frac{1}{p(x)} = \sum_x \sum_j \left(p_{\mathcal{A}}\left(I_j\right) p_{\mathcal{A}}\left(x|I_j\right)\log\frac{1}{p(x)}\right) \tag{7}$$

$$= \sum_j p_{\mathcal{A}}(I_j)\big(\mathbb{E}_{X \sim p_{\mathcal{A}}(x|I_j)}\left[-\log p\left(X\right)\right]\big).$$

where (7) follows from (6). $\qquad\square$

We will choose the intervals and algorithm $\mathcal{A}$ appropriately. By estimating each term in the summation above, we will design an algorithm with $T$ intervals that uses $O\left(\frac{k(\log^{(T)} k + \log(1/\varepsilon))^2}{\varepsilon^3}\right)$ samples and a constant words of space, and estimates $H(p)$ to $\pm\varepsilon$.

In Section 3.1, we provide the details with $T = 2$. This section will flesh out the key arguments, and finally in Section 3.2, we extend this to $T = \log^* k$ where $\log^* k = \min_i\{\log^{(i)} k \leq 1\}$ intervals to further reduce the sample complexity to $O(k(\log(1/\varepsilon))^2/\varepsilon^3)$.

## 3.1 Two Intervals Algorithm

We propose Algorithm 5 with $T = 2$ and the following guarantee.

**Theorem 8.** *Algorithm 5 uses* $O(NR + N_1R_1 + N_2R_2) = O\left(\frac{k(\log(\log(k)/\varepsilon))^2}{\varepsilon^3}\right)$ *samples,* 20 *words and outputs an* $\pm\varepsilon$ *estimate of* $H(p)$ *with probability at least* $2/3$.

### 3.1.1 Description of the Algorithm

Let $T = 2$, and $\beta > 16$ be a constant. Consider the following partition of $[0, 1]$:

$$I_2 = [0, \ell)\,, I_1 = [\ell, 1] \quad \text{where} \quad \ell = (\log k)^\beta/k. \tag{8}$$

We now specify the algorithm $\mathcal{A} : [k] \to \{I_1, I_2\}$ to be used in Lemma 7. $\mathcal{A}$ is denoted by ESTINT (Algorithm 2). For $x \in [k]$, it takes $N$ samples from $p$, and outputs the interval where the empirical fraction of occurrences of $x$ lies. ESTINT tries to predict the interval in which $p(x)$ lies.

| **Algorithm 2** $\mathcal{A}$ : ESTINT $(N, x)$ | **Algorithm 3** ESTPROBINT $(N, R)$ |
|---|---|
| 1: Obtain $N$ samples from $p$ | 1: $\hat{p}_{\mathcal{A}}\left(I_1\right) = 0$ |
| 2: **if** $x$ appears $\geq N\ell$ times, output $I_1$ | 2: **for** $t = 1$ to $R$ **do** |
| 3: **else** output $I_2$ | 3: $\quad$ Sample $x \sim p$. |
| | 4: $\quad$ **if** ESTINT $(N, x) = I_1$ **then** |
| | 5: $\qquad \hat{p}_{\mathcal{A}}\left(I_1\right) = \hat{p}_{\mathcal{A}}\left(I_1\right) + 1/R$ |

By Lemma 7, $H\left(p\right) = p_{\mathcal{A}}\left(I_1\right)H_1 + p_{\mathcal{A}}\left(I_2\right)H_2$. We estimate the terms in this expression as follows.

**Estimating $p_{\mathcal{A}}(I_j)$'s.** We run ESTINT multiple times on samples generated from $p$, and output the fraction of times the output is $I_j$ as an estimate of $p_{\mathcal{A}}(I_j)$. We only estimate $p_{\mathcal{A}}(I_1)$, since $p_{\mathcal{A}}(I_1) + p_{\mathcal{A}}(I_2) = 1$. The complete procedure is specified in Algorithm 3.

**Estimating $H_j$'s.** Recall that $H_j$'s are the expectations of $-\log\left(p(x)\right)$ under different distributions given in (6). Since the expectations are with respect to the conditional distributions, we first sample a symbol from $p$ and then conditioned on the event that ESTINT outputs $I_j$, we use an algorithm similar to Algorithm 1 to estimate $\log(1/p(x))$. The full algorithm is in Algorithm 4. Notice that when computing $\hat{H}_2$ in Step 8, we clip the $\hat{H}_2$'s to $\log \frac{1}{4\ell}$ if $N_{x,2} > 4\ell N_2 - 1$. This is done to restrict each $\hat{H}_2$ to be in the range of $[\log \frac{1}{4\ell}, \log N_2]$, which helps when proving concentration.

### 3.1.2 Performance Guarantees

**Memory Requirements.** We reserve 5 words to store parameters $R_1, R_2, N_1, N_2$ and $\ell$. ESTINT uses one word to keep track of the number of occurrences of $x$. For ESTPROBINT, we use one word to store $x$ and one word to keep track of the final sum $\hat{p}_{\mathcal{A}}\left(I_1\right)$. We execute CONDEXP for each interval separately and use one word each to store $x$ and keep track of $S_i$ and $\hat{H}_i$. We use two words to store the outputs $\bar{H}_1$ and $\bar{H}_2$ and store the final output $\hat{H}_{II}$ in one of those. Hence, at most 20 words of memory are sufficient.

**Algorithm 4** Estimating $H_1$ and $H_2$ : $\text{CONDEXP}\,(N_1, N_2, R_1, R_2)$

1: **for** $i = 1, 2$, set $\hat{H}_i = 0, S_i = 0$, **do**
2:     **for** $t = 1$ to $R_i$ **do**
3:         Sample $x \sim p$
4:         **if** $\text{ESTINT}\,(N, x) = I_i$ **then**
5:             $S_i = S_i + 1$
6:             Let $N_{x,i} \leftarrow$ # occurrences of $x$ in the next $N_i$ samples
7:             $\hat{H}_i = \hat{H}_i + \log\left(N_i/(N_{x,i}+1)\right)$ if $i = 1$
8:             $\hat{H}_i = \hat{H}_i + \max\left\{\log\left(N_i/(N_{x,i}+1)\right), \log\left(1/4\ell\right)\right\}$ if $i = 2$
9:     $\bar{H}_i = \hat{H}_i/S_i$

---

**Algorithm 5** Entropy Estimation with constant space: Two Intervals Algorithm

**Require:** Accuracy parameter $\varepsilon > 0, \gamma = \beta/2$, a data stream $X_1, X_2, \ldots \sim p$
1: Set
$$N = N_1 = \frac{C_1 k}{\varepsilon\,(\log k)^\gamma}, \ R = R_1 = C_2 \frac{\log(k/\varepsilon)^2}{\varepsilon^2}, \ N_2 = C_1 \cdot \frac{k}{\varepsilon}, \ R_2 = C_2 \cdot \frac{\left(\log((\log k)/\varepsilon)\right)^2}{\varepsilon^2}$$

2: $\hat{p}_{\mathcal{A}}\,(I_1) = \text{ESTPROBINT}\,(N, R)$
3: $\bar{H}_1, \bar{H}_2 = \text{CONDEXP}\,(N_1, N_2, R_1, R_2)$
4: $\hat{H}_{II} = \hat{p}_{\mathcal{A}}\,(I_1)\,\bar{H}_1 + (1 - \hat{p}_{\mathcal{A}}\,(I_1))\bar{H}_2$

---

**Sample Guarantees.** Let $\hat{H}_{II}^*$ be the unclipped version of the estimator where we don't use clipping in Step 8 in Algorithm 4 (all other steps remain the same). Then we can bound the estimation error by the following three terms and we will bound each of them separately,

$$\left|H\,(p) - \hat{H}_{II}\right| \leq \underbrace{\left|H\,(p) - \mathbb{E}\left[\hat{H}_{II}^*\right]\right|}_{\text{Unclipped Bias}} + \underbrace{\left|\mathbb{E}\left[\hat{H}_{II}\right] - \mathbb{E}\left[\hat{H}_{II}^*\right]\right|}_{\text{Clipping Error}} + \underbrace{\left|\hat{H}_{II} - \mathbb{E}\left[\hat{H}_{II}\right]\right|}_{\text{Concentration}}.$$

**Clipping Error.** By the design of $\text{CONDEXP}$, $\hat{H}_2$ is clipped only when the event $\mathcal{E}_x = \{\text{ESTINT}(N, x) = I_2, N_{x,2} > 4N_2\ell - 1\}$ occurs for some $x \in \mathcal{X}$. We bound the clipping error in the following lemma (proof in Section D.3) by showing that $\Pr\,(\mathcal{E}_x)$ is small.

**Lemma 9.** *(**Clipping Error Bound**) Let $\hat{H}_{II}$ be the entropy estimate of Algorithm 5 and let $\hat{H}_{II}^*$ be the entropy estimate of the unclipped version of Algorithm 5. Then $\left|\mathbb{E}\left[\hat{H}_{II}\right] - \mathbb{E}\left[\hat{H}_{II}^*\right]\right| \leq \varepsilon/3$.*

**Concentration Bound.** To prove the concentration bound, we use Lemma 10 to decompose it into three terms each of which can be viewed as the difference between some empirical mean and its true expectation which can be bounded using concentration inequalities. (proof in Section D.4)

**Lemma 10.** *(**Concentration Bound**) Let $\hat{H}_{II}$ be the entropy estimate of Algorithm 5 and let $\bar{H}_i$ be as defined in Algorithm 5. Let $p_{\mathcal{A}}\,(I_i)$ be the distribution defined in (6) where $\mathcal{A}$ is $\text{ESTINT}$.*

$$\left|\mathbb{E}\left[\hat{H}_{II}\right] - \hat{H}_{II}\right| \leq \sum_{i=1}^{2} p_{\mathcal{A}}\,(I_i)\left|\bar{H}_i - \mathbb{E}\left[\bar{H}_i\right]\right| + |p_{\mathcal{A}}\,(I_1) - \hat{p}_{\mathcal{A}}\,(I_1)|\,|\bar{H}_1 - \bar{H}_2| \leq \varepsilon/3.$$

We provide a brief outline of the proof below. By union bound, in order to show that with probability at least $2/3$ the sum is less than $\varepsilon/3$, it is sufficient to show that with probability at most $\frac{1}{9}$, each of the terms is greater than $\varepsilon/9$.

To bound $|p_{\mathcal{A}}\,(I_1) - \hat{p}_{\mathcal{A}}\,(I_1)|\,|\bar{H}_1 - \bar{H}_2|$, we first bound the range of $|\bar{H}_1 - \bar{H}_2|$ and then use Hoeffding's inequality (Lemma 2) to obtain concentration of $\hat{p}_{\mathcal{A}}\,(I_1)$. To bound $\left|\bar{H}_i - \mathbb{E}\left[\bar{H}_i\right]\right|$, note that we cannot obtain concentration using Hoeffding's inequality because $R_i$ (the number of samples that we average over) is a random variable. Therefore we apply Random Hoeffding Inequality (Lemma 3) to $\bar{H}_i$. Since $R_i$ depends on the range of the random variables being averaged over, we obtain a reduction in the sample complexity for $i = 2$ because of clipping the estimate below

to $\log \frac{1}{4\ell}$. Therefore the range for the second interval is $\log(N_2) - \log\frac{1}{4\ell} = O\left(\log\left((\log k)/\varepsilon\right)\right)$ implying $R_2 = O\left(\left(\log\left((\log k)/\varepsilon\right)\right)^2/\varepsilon^2\right)$ suffices for the desired probability. For $i = 1$, since the range is the same as the one interval case, we use the same $R_1$ as in the one interval case. Note $R_2 < R_1$.

**Bias Bound.** We bound the bias of the unclipped version, $\hat{H}_{II}^*$ using the following lemma:

**Lemma 11.** *(Unclipped Bias Bound) Let $\hat{H}_{II}^*$ be the unclipped estimate of Algorithm 5 and let $p_{\mathcal{A}}(I_i|x)$ be the conditional distribution defined in (6) where $\mathcal{A}$ is* ESTPROBINT. *Then,*

$$\left| H(p) - \mathbb{E}\left[\hat{H}_{II}^*\right] \right| \leq \sum_{i=1}^{2} \left( \sum_{x \in \mathcal{X}} p_{\mathcal{A}}(I_i|x)/N_i \right) \leq \varepsilon/3. \tag{9}$$

(Proof in Section D.2) Lemma 11 allows us to choose $N_1$ and $N_2$ separately to bound the bias. Interval $I_2$'s contribution is at most $\frac{k}{N_2}$. For interval $I_1$, we improve upon $\frac{k}{N_1}$ by partitioning $\mathcal{X}$ into sets $\mathcal{X}_1 = \{x \in \mathcal{X} | p(x) < \ell/2\}$ and $\mathcal{X}_2 = \{x \in \mathcal{X} | p(x) \geq \ell/2\}$. For $\mathcal{X}_1$, $p_{\mathcal{A}}(I_1|x)$ is small by Chernoff bound. For $\mathcal{X}_2$, since $p(x) \geq \ell/2$, $|\mathcal{X}_2| \leq 2/\ell$ which is smaller than $k$. Hence we can choose $N_2 < N_1$.

In the sample complexity of the two interval algorithm, observe that the term $N_2 R_2$ dominates. Reducing $N_2$ is hard because it is independent of the interval length. Therefore we hope to reduce $R_2$ by partitioning into intervals with smaller lengths. In the smallest interval, if we reduce the range of each estimate to be within a constant, then $O(\frac{1}{\varepsilon^2})$ samples would suffice for concentration. In the next section, we make this concrete by considering an algorithm that uses multiple intervals.

## 3.2 General Intervals Algorithm

The general algorithm follows the same principles as the previous section with a larger number of intervals, decreasing the sample requirements at each step, as discussed in Section 1.3. However, the proofs are much more involved, particularly in order to obtain an $O(k)$ upper bound on the sample complexity. We will sketch some of the key points and move a bulk of the algorithm and details to the appendix due to lack of space.

**Intervals.** Let $T = \log^* k$, where $\log^* k := \min_i \{\log^{(i)} k \leq 1\}$. Consider the following partition of $[0,1]$: $\{I_i\}_{i=1}^{T}$ where $I_1 = [l_1, h_1]$ and for $i = 2, ..., T$, $I_i = [l_i, h_i)$, $h_i = \frac{(\log^{(i-1)}(k))^\beta}{k}$ $(\beta > 16)$ and $\ell_{i-1} = h_i$. Define $l_T = 0$ and $h_1 = 1$, then we have for $i = 2, ..., T - 1$ :

$$I_1 = \left[ \frac{(\log^{(1)}(k))^\beta}{k}, 1 \right], I_T = \left[ 0, \frac{(\log^{(T-1)}(k))^\beta}{k} \right), I_i = \left[ \frac{(\log^{(i)}(k))^\beta}{k}, \frac{(\log^{(i-1)}(k))^\beta}{k} \right).$$

We divide the bottleneck of the two intervals algorithm $I_2$, into further intervals until the width of the smallest interval is a constant over $k$ ($e^\beta/k$) which implies concentration with lesser samples than before. Using Lemma 7, similar to the two intervals case, we will estimate each of the $p_{\mathcal{A}}(I_i)$ and $H_i$'s independently in Algorithm 8 (GENESTPROBINT) and Algorithm 9 (GENCONDEXP), presented in Appendix E.1. Complete algorithm for $T = \log^* k$ is presented in Algorithm 6.

**Memory Requirements.** The analysis of memory requirement is similar to that of the two interval case. To store parameters $\ell_i, N_i, R_i$'s, we only store $k, \varepsilon, \gamma, C_N$ and $C_R$ and compute the parameters on the fly. Notice that for each interval, the execution of GENESTINT, GENESTPROBINT and GENCONDEXP require same memory as that of their two interval counterparts. The trick here is that we don't need to store $\hat{p}_{\mathcal{A}}(I_i)$'s and $\bar{H}_i$'s since we can perform each of GENESTPROBINT and GENCONDEXP for one interval and maintain a running sum of $\hat{p}_{\mathcal{A}}(I_i)\bar{H}_i$'s. Therefore, Algorithm 6 uses at most 20 words of space.

**Sample complexity.** Algorithm 6 proves the main claim of our paper in Theorem 1. The key idea to remove the extra loglog factor in Theorem 8 is to progressively make the error requirements stricter for the larger probability intervals. We denote the final estimate without the clipping step (Step 8) by $\hat{H}_{\mathcal{I}}^*$ (all other steps remaining the same). Then the error can be bounded by the following three terms:

$$|H(p) - \hat{H}_{\mathcal{I}}| \leq \underbrace{|H(p) - \mathbb{E}\left[\hat{H}_{\mathcal{I}}^*\right]|}_{\text{Unclipped Bias}} + \underbrace{|\mathbb{E}\left[\hat{H}_{\mathcal{I}}\right] - \mathbb{E}\left[\hat{H}_{\mathcal{I}}^*\right]|}_{\text{Clipping Error}} + \underbrace{|\hat{H}_{\mathcal{I}} - \mathbb{E}\left[\hat{H}_{\mathcal{I}}\right]|}_{\text{Concentration}}. \tag{10}$$

---

**Algorithm 6** Entropy Estimation with constant space: General Intervals Algorithm

---

**Require:** Accuracy parameter $\varepsilon > 0, \gamma = \beta/2$, a data stream $X_1, X_2, \ldots \sim p$.

1: Set
$$N_i = C_N \cdot \frac{k}{\varepsilon(\log^{(i)}(k))^\gamma}, \qquad R_i = C_R \cdot \frac{(\log(\log^{(i-1)}(k)/\varepsilon))^2}{\varepsilon^2} \qquad 1 \le i \le T-1$$

$$N_T = C_N \cdot \frac{k}{\varepsilon}, \qquad R_T = C_R \cdot \frac{(\log(\log^{(T-1)}(k)/\varepsilon))^2}{\varepsilon^2}$$

2: $\{\hat{p}_{\mathcal{A}}(I_i)\}_{i=1}^{T-1} = \text{GENESTPROBINT}\left(\{N_i\}_{i=1}^{T-1}, \{R_i\}_{i=1}^{T-1}\right)$

3: $\{\bar{H}_i\}_{i=1}^{T} = \text{GENCONDEXP}\left(\{N_i\}_{i=1}^{T}, \{R_i\}_{i=1}^{T}\right)$

4: $\hat{H}_{\mathcal{I}} = \sum_{i=1}^{T-1} \hat{p}_{\mathcal{A}}(I_i) \bar{H}_i + (1 - \sum_{i=1}^{T-1} \hat{p}_{\mathcal{A}}(I_i))\bar{H}_T$

---

With the parameters defined in Algorithm 6, we can bound the unclipped bias and clipping error in (10) by $\frac{\varepsilon}{3}$ each and show that the concentration part is also bounded by $\frac{\varepsilon}{3}$ with probability at least $2/3$. The details are given in Lemma 13, 14, and 15 in Appendix E.

## 4 Open Problems

There are several interesting questions that arise from our work. While our algorithms require only a constant memory words of space, they require a $\log k$ multiplicative factor more samples (as a function of $k$) than the optimal sample complexity (in (3)).

- Does there exist an algorithm for entropy estimation that has the optimal sample complexity and space requirement that is at most $\text{poly}(\log k)$?

We are unaware of any implementation that requires sub-linear space in $k$. Designing a strictly sublinear-space (space requirement $k^\alpha$ for some $\alpha < 1$) sample-optimal algorithm could be a first step toward solving the question above. At the same time, there might not exist an algorithm with a small sample complexity. This leads to the following complementary question.

- Prove a lower bound on the space requirement of a sample-optimal algorithm for entropy estimation.

Beyond these, obtaining sample-space trade-offs for distribution testing, and property estimation tasks is an exciting future direction.

**Acknowledgements.** This work is supported by NSF-CCF-1657471. This research started with the support of MIT-Shell Energy Research Fellowship to JA and PI, while JA was at MIT.

## Footnotes

[1]We use space, storage, and memory interchangeably.

[2]For smaller $\delta$'s, we can apply median trick with an extra factor of $\log(1/\delta)$ samples.

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
