[Supplementary Material · neurips2019_submission.pdf]

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

## A A Bound on Expression of Binomial Random Variables

**Lemma 12.** *Let $X \sim Bin\,(m, r)$, then $\mathbb{E}\left[\frac{1}{X+1}\right] \leq \frac{1}{r(m+1)}$.*

*Proof.*

$$\mathbb{E}\left[\frac{1}{X+1}\right] = \frac{1}{m+1} \sum_{l=0}^{m} \frac{m+1}{l+1}\binom{m}{l} r^l (1-r)^{m-l} = \frac{1-(1-r)^{m+1}}{r\,(m+1)} \leq \frac{1}{r\,(m+1)}. \qquad \square$$

## B Proof of Random Hoeffding Inequality (Lemma 3)

First, observe that $|X - \mathbb{E}[X]| \leq b - a$. Hence if $t > p(b-a)$, the left hand side is zero and the inequality naturally holds. Next, we assume $t \leq p(b-a)$, which is equivalent to $p \geq \frac{t^2}{p(b-a)^2}$.

$$\Pr\left(|X - \mathbb{E}[X]| \geq \frac{t}{p}\right) = \sum_{r=0}^{m} \Pr\left(|X - \mathbb{E}[X]| \geq \frac{t}{p}\bigg| M = r\right)\Pr(M = r).$$

We can divide the above into two parts, $r \leq \left\lfloor\frac{mp}{2}\right\rfloor$ and $r \geq \left\lceil\frac{mp}{2}\right\rceil$. For the first part, by Chernoff bound, we get

$$\sum_{r=0}^{\left\lfloor\frac{mp}{2}\right\rfloor} \Pr\left(|X - \mathbb{E}[X]| \geq \frac{t}{p}\bigg| M = r\right)\Pr(M = r) \leq \Pr\left(M \leq \frac{mp}{2}\right) \leq \exp\left(\frac{-mp}{8}\right).$$

For the second part, by Hoeffding Inequality (Lemma 2), we have:

$$\sum_{r=\left\lceil\frac{mp}{2}\right\rceil}^{m} 2\exp\left(\frac{-2rt^2}{p^2\,(b-a)^2}\right)\Pr(M = r) \leq 2\exp\left(\frac{-2t^2}{p^2\,(b-a)^2}\frac{mp}{2}\right) \leq 2\exp\left(\frac{-mt^2}{p\,(b-a)^2}\right).$$

Combining the two, we get:

$$\Pr\left(|X - \mathbb{E}[X]| \geq \frac{t}{p}\right) \leq \exp\left(\frac{-mp}{8}\right) + 2\exp\left(\frac{-mt^2}{p\,(b-a)^2}\right) \leq 3\exp\left(\frac{-mt^2}{8p\,(b-a)^2}\right).$$

## C Proofs from Section 2

### C.1 Proof of Lemma 5 : Bias Bound

From Algorithm 1, we can express $\bar{H}$ as

$$\bar{H} = \frac{1}{R}\sum_{t=1}^{R}\sum_{x\in\mathcal{X}}\mathbb{1}\{x = x_t\}\log\left(\frac{N}{N_{x_t}+1}\right).$$

$$= \sum_{x\in\mathcal{X}}\frac{1}{R}\sum_{t=1}^{R}\mathbb{1}\{x = x_t\}\log\left(\frac{N}{N_{x_t}+1}\right).$$

The above formulation can be thought of as an empirical average of $\log\left(\frac{N}{N_X+1}\right)$, where $X \sim p$. Therefore,

$$\mathbb{E}\left[\bar{H}\right] = \sum_{x\in\mathcal{X}}p\,(x)\,\mathbb{E}\left[\log\left(\frac{N}{N_x+1}\right)\right]. \tag{11}$$

$$H(p) - \mathbb{E}\left[\bar{H}\right] = \sum_{x \in \mathcal{X}} p(x) \log \frac{1}{p(x)} - \sum_{x \in \mathcal{X}} p(x) \mathbb{E}\left[\log\left(\frac{N}{N_x + 1}\right)\right]$$

$$= \sum_{x \in \mathcal{X}} p(x) \mathbb{E}\left[\log\left(\frac{N_x + 1}{Np(x)}\right)\right]$$

$$\leq \sum_{x \in \mathcal{X}} p(x) \log\left(\mathbb{E}\left[\frac{N_x + 1}{Np(x)}\right]\right) \tag{12}$$

$$= \sum_{x \in \mathcal{X}} p(x) \log\left(1 + \frac{1}{Np(x)}\right)$$

$$\leq \frac{k}{N}. \tag{13}$$

where we obtain (12) using Jensen's inequality and (13) follows from $\log(1 + x) \leq x$. We now bound $\mathbb{E}\left[\bar{H}\right] - H(p)$.

$$\mathbb{E}\left[\bar{H}\right] - H(p) = \sum_{x \in \mathcal{X}} p(x) \mathbb{E}\left[\log\left(\frac{Np(x)}{N_x + 1}\right)\right]$$

$$\leq \sum_{x \in \mathcal{X}} p(x) \log\left(\mathbb{E}\left[\frac{Np(x)}{N_x + 1}\right]\right) \tag{14}$$

$$\leq \sum_{x \in \mathcal{X}} p(x) \log\left(\frac{N}{N + 1}\right) < 0. \tag{15}$$

where (14) is obtained using Jensen's inequality and (15) follows from Lemma 12. Therefore,

$$\left|\mathbb{E}\left[\bar{H}\right] - H(p)\right| \leq \frac{k}{N}. \tag{16}$$

## C.2 Proof of Lemma 6 : Concentration Bound

Note that $\bar{H}$ is an average of $R$ i.i.d random variables

$$Z_t = \sum_{x \in \mathcal{X}} \mathbb{1}\left\{x = x_t\right\} \log\left(\frac{N}{N_{x_t} + 1}\right).$$

where $t \in [R]$. Each of the $R$ random variables can take values from $\left[\log\left(\frac{N}{N+1}\right), \log N\right]$. Applying Hoeffding's inequality (Lemma 2),

$$\Pr\left(\left|\bar{H} - \mathbb{E}\left[\bar{H}\right]\right| \geq \frac{\varepsilon}{2}\right) \leq \exp\left(-\frac{R\varepsilon^2}{2\log^2(N + 1)}\right). \tag{17}$$

# D   Two interval Algorithm Proofs

## D.1   Expectation of Unclipped Version Estimates

Let $S_i$ be the number of times ESTINT $= I_i$ during the $R_i$ iterations for interval $I_i$ in CONDEXP. Let $S_{x,i}$ be the number of times symbol $x$ is the first sampled element among these. Note that $S_i \sim \text{Bin}(R_i, p_{\mathcal{A}}(I_i))$ and $S_{x,i} \sim \text{Bin}(S_i, p_{\mathcal{A}}(x \mid I_i))$ where $p_{\mathcal{A}}(x \mid I_i) = \dfrac{p(x)\, p_{\mathcal{A}}(I_i \mid x)}{p_{\mathcal{A}}(I_i)}$. Let $N_{x,i,v}$ be $N_{x,i}$ (defined in CONDEXP) when $x$ is sampled and ESTINT$(N, x) = I_i$ for the $v^{\text{th}}$ time. Denote the unclipped version of $\bar{H}_i$ by $\bar{H}_i^*$. We can write $\bar{H}_i^*$ as follows

$$\bar{H}_i^* = \frac{1}{S_i} \sum_{x \in \mathcal{X}} \sum_{v=1}^{S_{x,i}} \log\left(\frac{N_i}{N_{x,i,v} + 1}\right). \tag{18}$$

The above equation implies that $\bar{H}_i^*$ is an empirical mean of $\log\left(\frac{N_i}{N_{x,i}+1}\right)$ where $X \sim p_{\mathcal{A}}(x \mid I_i)$. Note that for a fixed $x$, $S_{x,i} \sim \text{Bin}(S_i, p_{\mathcal{A}}(x|I_i))$. Therefore, the expectation is

$$\mathbb{E}\left[\bar{H}_i^*\right] = \sum_{x \in \mathcal{X}} \frac{p(x) p_{\mathcal{A}}(I_i|x)}{p_{\mathcal{A}}(I_i)} \mathbb{E}\left[\log\left(\frac{N_i}{N_{x,i}+1}\right)\right]. \tag{19}$$

### D.2 Proof of Lemma 11 : Unclipped Bias Bound

Define $\bar{H}_1^*$ and $\bar{H}_2^*$ to be the analog of $\bar{H}_1$ and $\bar{H}_2$ in the unclipped version of Algorithm 5. We first note that

$$\mathbb{E}\left[\hat{H}_{II}^*\right] = p_{\mathcal{A}}(I_1)\mathbb{E}\left[\bar{H}_1^*\right] + (1 - p_{\mathcal{A}}(I_1))\mathbb{E}\left[\bar{H}_2^*\right]. \tag{20}$$

The above is true since, $\mathbb{E}[\hat{p}_{\mathcal{A}}(I_1)] = p_{\mathcal{A}}(I_1)$ and Algorithm 4 estimates $\hat{p}_{\mathcal{A}}(I_1)$ and $\bar{H}_1^*, \bar{H}_2^*$ independently.

We use the following result from equation 19 in Section D.1

$$\mathbb{E}\left[\bar{H}_i^*\right] = \sum_{x \in \mathcal{X}} \frac{p(x) p_{\mathcal{A}}(I_i|x)}{p_{\mathcal{A}}(I_i)} \mathbb{E}\left[\log\left(\frac{N_i}{N_{x,i}+1}\right)\right]. \tag{21}$$

Using Lemma (7) and Jensen's inequality, we have

$$\mathbb{E}\left[\hat{H}_{II}^*\right] - H(p) \leq \sum_{i=1}^{2} p_{\mathcal{A}}(I_i) \left(\sum_{x \in \mathcal{X}} \frac{p(x) p_{\mathcal{A}}(I_i|x)}{p_{\mathcal{A}}(I_i)} \mathbb{E}\left[\log\left(\frac{N_i p(x)}{N_{x,i}+1}\right)\right]\right)$$

$$\leq \sum_{i=1}^{2} p_{\mathcal{A}}(I_i) \left(\sum_{x \in \mathcal{X}} \frac{p(x) p_{\mathcal{A}}(I_i|x)}{p_{\mathcal{A}}(I_i)} \log\left(\mathbb{E}\left[\frac{N_i p(x)}{N_{x,i}+1}\right]\right)\right)$$

$$\leq \sum_{i=1}^{2} p_{\mathcal{A}}(I_i) \left(\sum_{x \in \mathcal{X}} \frac{p(x) p_{\mathcal{A}}(I_i|x)}{p_{\mathcal{A}}(I_i)} \log\left(\frac{N_i}{N_i+1}\right)\right) \leq 0. \tag{22}$$

where (22) follows from Lemma 12. To bound the reverse, using Lemma (7), Jensen's inequality and the fact that $\log(1+x) \leq x$, we have

$$H(p) - \mathbb{E}\left[\hat{H}_{II}^*\right] = \sum_{i=1}^{2} p_{\mathcal{A}}(I_i) \left(\sum_{x \in \mathcal{X}} \frac{p(x) p_{\mathcal{A}}(I_i|x)}{p_{\mathcal{A}}(I_i)} \mathbb{E}\left[\log\left(\frac{N_{x,i}+1}{N_i p(x)}\right)\right]\right)$$

$$\leq \sum_{i=1}^{2} p_{\mathcal{A}}(I_i) \left(\sum_{x \in \mathcal{X}} \frac{p(x) p_{\mathcal{A}}(I_i|x)}{p_{\mathcal{A}}(I_i)} \log\left(\mathbb{E}\left[\frac{N_{x,i}+1}{N_i p(x)}\right]\right)\right)$$

$$= \sum_{i=1}^{2} p_{\mathcal{A}}(I_i) \left(\sum_{x \in \mathcal{X}} \frac{p(x) p_{\mathcal{A}}(I_i|x)}{p_{\mathcal{A}}(I_i)} \log\left(\frac{N_i p(x)+1}{N_i p(x)}\right)\right)$$

$$\leq \sum_{i=1}^{2} p_{\mathcal{A}}(I_i) \left(\sum_{x \in \mathcal{X}} \frac{p_{\mathcal{A}}(I_i|x)}{N_i p_{\mathcal{A}}(I_i)}\right)$$

$$= \sum_{i=1}^{2} \left(\sum_{x \in \mathcal{X}} \frac{p_{\mathcal{A}}(I_i|x)}{N_i}\right). \tag{23}$$

For interval $I_1$, we partition $\mathcal{X}$ into two sets $\mathcal{X}_1 = \{x \in \mathcal{X}|p(x) < \ell/2\}$ and $\mathcal{X}_2 = \{x \in \mathcal{X}|p(x) \geq \ell/2\}$. For $x \in \mathcal{X}_1$, the probability that algorithm $\text{EstInt}(N, x) = I_1$ is small. In particular, by Chernoff bound,

$$p_{\mathcal{A}}(I_1|x) = \Pr(N_x > N_1\ell) \leq \exp\left(-\frac{N_1\ell}{6}\right). \tag{24}$$

For $x \in \mathcal{X}_2$, since $p(x) \geq \ell/2$, $|\mathcal{X}_2| \leq \frac{2}{\ell}$ and each $p_{\mathcal{A}}(I_1|x) \leq 1$, we have

$$\sum_{x \in \mathcal{X}} \frac{p_{\mathcal{A}}(I_1|x)}{N_1} = \sum_{x \in \mathcal{X}_1} \frac{p_{\mathcal{A}}(I_1|x)}{N_1} + \sum_{x \in \mathcal{X}_2} \frac{p_{\mathcal{A}}(I_1|x)}{N_1} \leq \frac{k}{N_1} \exp\left(-\frac{N_1 \ell}{6}\right) + \frac{2}{N_1 \ell}. \qquad (25)$$

For interval $I_2$, we simply bound each term by 1 and get

$$\sum_{x \in \mathcal{X}} \frac{p_{\mathcal{A}}(I_2|x)}{N_2} \leq \frac{k}{N_2}. \qquad (26)$$

Plugging in the values of $N_1$, $N_2$ defined in Algorithm 5, it is easy to see there exists a constant $C_1$ such that (26) and (25) are bounded above by $\frac{\varepsilon}{6}$ which completes the proof.

### D.3 Proof of Lemma 9 : Clipping Error Bound

Define $\bar{H}_1^*$ and $\bar{H}_2^*$ to be the analog of $\bar{H}_1$ and $\bar{H}_2$ in the unclipped version of Algorithm 4. Using (20) and the fact that the clipping step is applied only when computing $\bar{H}_2$, we have

$$\left| \mathbb{E}\left[\hat{H}_{II}\right] - \mathbb{E}\left[\hat{H}_{II}^*\right] \right| \leq p_{\mathcal{A}}(I_2) \left| \mathbb{E}\left[\bar{H}_2\right] - \mathbb{E}\left[\bar{H}_2^*\right] \right|. \qquad (27)$$

From Algorithm 5, we note that $\bar{H}_2$ is different from $\bar{H}_2^*$ only when $\mathcal{E}_x = \{\textsc{EstInt}(N, x) = I_2, N_{x,2} > 4N_2 \ell - 1\}$ occurs. Therefore from (19), we have the following

$$\left| \mathbb{E}\left[\bar{H}_2\right] - \mathbb{E}\left[\bar{H}_2^*\right] \right| \leq \sum_{x \in \mathcal{X}} \Pr\left(N_{x,2} > 4N_2 \ell - 1\right) \frac{p(x) p_{\mathcal{A}}(I_2|x)}{p_{\mathcal{A}}(I_2)} \left(\log\left(\frac{1}{4\ell}\right) - \log\left(\frac{N_2}{N_2 + 1}\right)\right)$$

$$\leq \sum_{x \in \mathcal{X}} \Pr\left(N_{x,2} > 4N_2 \ell - 1\right) \frac{p(x) p_{\mathcal{A}}(I_2|x)}{p_{\mathcal{A}}(I_2)} \log k. \qquad (28)$$

By Chernoff bound, if $p(x) > 2\ell$

$$\Pr\left(\textsc{EstInt}(N, x) = I_2\right) \leq \exp\left(-\frac{N\ell}{3}\right).$$

And if $p(x) < 2\ell$,

$$\Pr\left(N_{x,2} > 4N_2 \ell - 1\right) = p_{\mathcal{A}}(I_2|x) \leq \exp\left(-\frac{2N_2 \ell}{3}\right).$$

Therefore, plugging in values of $N$ and $N_2$, we have

$$p_{\mathcal{A}}(I_2) \left| \mathbb{E}\left[\bar{H}_2\right] - \mathbb{E}\left[\bar{H}_2^*\right] \right| \leq \min\left\{\exp\left(-\frac{N\ell}{3}\right), \exp\left(-\frac{2N_2 \ell}{3}\right)\right\} \log k \leq \frac{\varepsilon}{3}. \qquad (29)$$

### D.4 Proof of Lemma 10 : Concentration Bound

Using (20), we have

$$\left| \mathbb{E}\left[\hat{H}_{II}\right] - \hat{H}_{II} \right| = \left| p_{\mathcal{A}}(I_1) \mathbb{E}\left[\bar{H}_1\right] + p_{\mathcal{A}}(I_2) \mathbb{E}\left[\bar{H}_2\right] - \hat{p}_{\mathcal{A}}(I_1) \bar{H}_1 - \hat{p}_{\mathcal{A}}(I_2) \bar{H}_2 \right|$$

$$= \left| p_{\mathcal{A}}(I_1) \mathbb{E}\left[\bar{H}_1\right] + p_{\mathcal{A}}(I_2) \mathbb{E}\left[\bar{H}_2\right] - p_{\mathcal{A}}(I_1) \bar{H}_1 - p_{\mathcal{A}}(I_2) \bar{H}_2 \right.$$

$$\left. + p_{\mathcal{A}}(I_1) \bar{H}_1 + p_{\mathcal{A}}(I_2) \bar{H}_2 - \hat{p}_{\mathcal{A}}(I_1) \bar{H}_1 - \hat{p}_{\mathcal{A}}(I_2) \bar{H}_2 \right|$$

$$\leq \sum_{i=1}^{2} \left| p_{\mathcal{A}}(I_i) \left(\mathbb{E}\left[\bar{H}_i\right] - \bar{H}_i\right) \right| + \left| \left(p_{\mathcal{A}}(I_1) - \hat{p}_{\mathcal{A}}(I_1)\right) \left(\bar{H}_1 - \bar{H}_2\right) \right|. \qquad (30)$$

(30) is true because by definition,

$$p_{\mathcal{A}}(I_1) - \hat{p}_{\mathcal{A}}(I_1) = -\left(p_{\mathcal{A}}(I_2) - \hat{p}_{\mathcal{A}}(I_2)\right)$$

We first bound $|p_{\mathcal{A}}(I_1) - \hat{p}_{\mathcal{A}}(I_1)||\bar{H}_1 - \bar{H}_2|$. Note that $\bar{H}_1 \in \left[\log \frac{N_1}{N_1+1}, \log N_1\right]$. And because of clipping, $\bar{H}_2 \in \left[\log \frac{N_2}{4N_2\ell+1}, \log N_2\right]$. Since $N_2 > N_1$, $\left|\hat{H}_1 - \hat{H}_2\right| \leq \log \frac{N_2(N_1+1)}{N_1}$. To bound $|p_{\mathcal{A}}(I_1) - \hat{p}_{\mathcal{A}}(I_1)|$, since $\hat{p}_{\mathcal{A}}(I_1)$ is the average of $R$ i.i.d binary random variables, by Hoeffding's inequality (Lemma 2) we have

$$\Pr\left(|p_{\mathcal{A}}(I_1) - \hat{p}_{\mathcal{A}}(I_1)| > t\right) \leq 2\exp\left(-2Rt^2\right).$$

Take $t = \frac{\varepsilon}{9\log \frac{N_2(N_1+1)}{N_1}}$. There exists constant $C_2$ such that for the value of $R_1$ from Algorithm 5, with probability at least $8/9$, we have:

$$|p_{\mathcal{A}}(I_1) - \hat{p}_{\mathcal{A}}(I_1)||\bar{H}_1 - \bar{H}_2| \leq \varepsilon/9.$$

To bound $\left|\bar{H}_i - \mathbb{E}\left[\bar{H}_i\right]\right|$ we cannot directly use Hoeffding's inequality because the number of samples that we are taking an average over is a random variable. We therefore apply the Random Hoeffding Inequality (Lemma 3) to $\bar{H}_1$ to get:

$$\Pr\left(p_{\mathcal{A}}(I_i)\left|\bar{H}_i - \mathbb{E}\left[\bar{H}_i\right]\right| > \varepsilon/9\right) \leq 3\exp\left(\frac{-R_i\varepsilon^2}{72p_{\mathcal{A}}(I_i)(b_i - a_i)^2}\right), \tag{31}$$

where $[a_i, b_i]$ is the possible range of each independent variables when estimating $\bar{H}_i$. Since $a_1 = \log\left(\frac{N_1}{N_1+1}\right), b_1 = \log(N_1), b_1 - a_1 = \log(N_1 + 1) = O(\log \frac{k}{\varepsilon})$. Therefore, there exists a constant $C_2$ such that $R_1 = C_2 \frac{\log^2(k/\varepsilon)}{\varepsilon^2}$ suffices to get a probability at least $8/9$.

The reduction in sample complexity is obtained for $i = 2$. Here $a_2 = \log \frac{1}{4\ell}$ instead of $\log\left(\frac{N_2}{N_2+1}\right)$ because of the clipping step. Since $b_2 = \log(N_2), b_2 - a_2 = \log(4N_2\ell) = O\left(\log\left((\log k)/\varepsilon\right)\right)$. Therefore, $\exists$ constant $C_2$, such that $R_2 = C_2 \cdot \frac{(\log((\log k)/\varepsilon))^2}{\varepsilon^2}$ would suffice to get a probability at least $8/9$.

# E    General Interval Algorithm

## E.1    General Interval Algorithms

We provide the pseudocode of the various procedures for our main algorithms in this section.

Algorithm 7 is the interval estimation algorithm that tests from samples where a $p(x)$ belongs to a certain interval $I_t$.

---

**Algorithm 7** Estimating intervals: General Case : GENESTINT($\{N_i\}_{i=1}^{t}, x$)

---

**Require:** $\{N_i\}_{i=1}^{t}, x$ drawn from $p$
1: **for** $i = 1$ to $t$ **do**
2:    Generate $N_i$ samples from $p$
3:    **if** $x$ appears more than $N_i\ell_i$ times **then** Output $I_i$
4: Output $I_T$

---

Algorithm 8 describes the procedure to estimate $p_{\mathcal{A}}(I_t)$'s.

Algorithm 9 estimates the entropy contributions from the intervals.

## E.2    Unclipped Bias Bound

In this part, we will bound the bias of the unclipped version of the entropy estimate. In particular, we will prove the following lemma:

**Lemma 13.** *(**Unclipped Bias bound**) Let $\hat{H}_{\mathcal{I}}^*$ be the entropy estimate given by Algorithm 6 without the clipping step in Algorithm 9 , then*

$$\left|\mathbb{E}\left[\hat{H}_{\mathcal{I}}^*\right] - H(p)\right| \leq \frac{\varepsilon}{3}. \tag{32}$$

**Algorithm 8** Estimating $p_\mathcal{A}(I_i)$, $1 \le i \le T-1$ : GENESTPROBINT($\{N_i\}_{i=1}^{T-1}, \{R_i\}_{i=1}^{T-1}$)

---

**Require:** $\{N_i\}_{i=1}^{T-1}, \{R_i\}_{i=1}^{T-1}$
1: **for** $i = 1, 2, ..., T-1$ **do**
2:     $\hat{p}_\mathcal{A}(I_i) = 0$
3:     **for** $t = 1$ to $R_i$ **do**
4:        Sample $x \sim p$.
5:        **if** GENESTINT$\left(\{N_j\}_{j=1}^{i}, x\right) = I_i$ **then** $\hat{p}_\mathcal{A}(I_i) = \hat{p}_\mathcal{A}(I_i) + \frac{1}{R_i}$

---

**Algorithm 9** Estimating $H_i$'s : GENCONDEXP$\left(\{N_i\}_{i=1}^{T}, \{R_i\}_{i=1}^{T}\right)$

---

**Require:** $\{N_i\}_{i=1}^{T}, \{R_i\}_{i=1}^{T}$
1: **for** $i = 1$ to $T$ **do**
2:     $\hat{H}_i = 0, S_i = 0$
3:     **for** $t = 1$ to $R_i$ **do**
4:        Generate $x \sim p$
5:        **if** GENESTINT$\left(\{N_j\}_{j=1}^{i}, x\right)$ is $I_i$ **then**
6:           $S_i = S_i + 1$
7:           Let $N_{x,i} \leftarrow$ # occurrences of $x$ in the next $N_i$ samples
8:           $E_{x,i} = \max\{\log\left(\frac{N_i}{N_{x,i}+1}\right), \log\frac{1}{4h_i}\}$
9:           $\hat{H}_i = \hat{H}_i + E_{x,i}$
10:     $\bar{H}_i = \frac{\hat{H}_i}{S_i}$

---

*Proof.* Denote the unclipped versions of $\bar{H}_i$ by $\bar{H}_i^*$. For interval $I_i$, let $S_i$ be the number of times GENESTINT$\left(\{N_j\}_{j=1}^{i}, x\right) = I_i$ during $R_i$ iterations in Algorithm 9. For $x \in \mathcal{X}$, let $S_{x,i}$ be the number of times symbol $x$ is the first sampled element among these. Note that $S_i \sim \text{Bin}(R_i, p_\mathcal{A}(I_i))$ and $S_{x,i} \sim \text{Bin}(S_i, p_\mathcal{A}(x \mid I_i))$. Let $N_{x,i,v}$ be $N_{x,i}$ (defined in GENCONDEXP) when $x$ is first sampled and GENESTINT$\left(\{N_j\}_{j=1}^{i}, x\right) = I_i$ for the $v^{\text{th}}$ time. We can write $\bar{H}_i^*$ as follows.

$$\bar{H}_i^* = \frac{1}{S_i}\sum_{x \in \mathcal{X}}\sum_{v=1}^{S_{x,i}}\log\left(\frac{N_i}{N_{x,i,v}+1}\right). \tag{33}$$

Since the computation of $\hat{p}_\mathcal{A}(I_i)$ and $\bar{H}_i^*$ is independent, we have

$$\mathbb{E}\left[\hat{H}_\mathcal{I}^*\right] = \sum_{i=1}^{T}\mathbb{E}\left[\hat{p}_\mathcal{A}(I_i)\right]\mathbb{E}\left[\bar{H}_i^*\right] = \sum_{i=1}^{T}p_\mathcal{A}(I_i)\mathbb{E}\left[\bar{H}_i^*\right]. \tag{34}$$

For the interval $I_i$, $\mathbb{E}\left[\bar{H}_i^*\right]$ can be written as (refer Section D.1 for detailed argument):

$$\mathbb{E}\left[\bar{H}_i^*\right] = \sum_{x \in \mathcal{X}}\frac{p(x)\, p_\mathcal{A}(I_i|x)}{p_\mathcal{A}(I_i)}\mathbb{E}\left[\log\left(\frac{N_i}{N_{x,i}+1}\right)\right].$$

Similar to Equations (22) and (23), we have

$$\mathbb{E}\left[\hat{H}_\mathcal{I}^*\right] - H(p) \le \sum_{i=1}^{T}p_\mathcal{A}(I_i)\left(\sum_{x \in \mathcal{X}}\frac{p(x)\, p_\mathcal{A}(I_i|x)}{p_\mathcal{A}(I_i)}\log\left(\frac{N_i}{N_i+1}\right)\right) \le 0.$$

$$H(p) - \mathbb{E}\left[\hat{H}_\mathcal{I}^*\right] \le \sum_{i=1}^{T}\left(\sum_{x \in \mathcal{X}}\frac{p_\mathcal{A}(I_i|x)}{N_i}\right).$$

Therefore,

$$\left|H(p) - \mathbb{E}\left[\hat{H}_\mathcal{I}^*\right]\right| \le \sum_{i=1}^{T}\left(\sum_{x \in \mathcal{X}}\frac{p_\mathcal{A}(I_i|x)}{N_i}\right). \tag{35}$$

For interval $I_i$, $1 \leq i \leq T - 1$, we divide the symbols into $\mathcal{X}_l = \left\{x : p_x \leq \frac{l_i}{2}\right\}$ and $\mathcal{X}_m = \left\{x : p_x > \frac{l_i}{2}\right\}$ and get

$$
\begin{aligned}
\sum_{x \in \mathcal{X}} \frac{p_{\mathcal{A}}\left(I_i | x\right)}{N_i} &\leq \sum_{x \in \mathcal{X}_l} \frac{p_{\mathcal{A}}\left(I_i | x\right)}{N_i} + \sum_{x \in \mathcal{X}_m} \frac{p_{\mathcal{A}}\left(I_i | x\right)}{N_i} \\
&\leq \frac{1}{N_i}\left(\sum_{x \in \mathcal{X}_l} \exp\left(-\frac{N_i l_i}{6}\right)\right) + \frac{2}{N_i l_i} \\
&\leq \frac{k}{N_i} \exp\left(-\frac{N_i l_i}{6}\right) + \frac{2}{N_i l_i}.
\end{aligned}
\tag{36}
$$

Substituting the values of $N_i, l_i$, we get that

$$
\begin{aligned}
\sum_{x \in \mathcal{X}} \frac{p_{\mathcal{A}}\left(I_i | x\right)}{N_i} &\leq \frac{\varepsilon}{C_N}\left(\log^{(i)} k\right)^\gamma \exp\left(-C_N \frac{\left(\log^{(i)} k\right)^{\beta - \gamma}}{6\varepsilon}\right) + 2\frac{\varepsilon}{C_N}\left(\log^{(i)} k\right)^{\gamma - \beta} \\
&\leq \frac{3}{C_N} \frac{\varepsilon}{\left(\log^{(i)} k\right)^\gamma}.
\end{aligned}
$$

The last inequality holds because $\beta = 2\gamma$ and $e^{-x} \leq \frac{1}{x^2}$ for $x > 0$. Hence we have:

$$
\sum_{i=1}^{T-1}\left(\sum_{x \in \mathcal{X}} \frac{p_{\mathcal{A}}\left(I_i | x\right)}{N_i}\right) \leq \frac{3\varepsilon}{C_N} \sum_{i=1}^{T-1} \frac{1}{\left(\log^{(i)} k\right)^\gamma} = \frac{3\varepsilon}{C_N} \sum_{i=1}^{T-1} \frac{1}{\left(\log^{(T-i)} k\right)^\gamma}.
\tag{37}
$$

Let $a_i = \log^{(T-i)} k$. Then $a_{i+1} = e^{a_i}$. Since $T = \log^* k$, we have $1 \leq a_1 \leq e$. It can be shown that $\frac{a_{i+1}}{a_i} = \frac{e^{a_i}}{a_i} \geq e$. Hence we get

$$
\forall i, a_i = \log^{(T-i)} k \geq e^{i-1}.
\tag{38}
$$

Therefore, we get:

$$
\sum_{i=1}^{T-1} \frac{1}{\left(\log^{(T-i)} k\right)^\gamma} = \sum_{i=1}^{T-1} \frac{1}{a_i^\gamma} \leq \sum_{i=1}^{T-1} \frac{1}{e^{\gamma(i-1)}} \leq 2.
$$

Plugging this in (37), we can see $\exists$ constant $C_N > 36$, such that:

$$
\sum_{i=1}^{T-1}\left(\sum_{x \in \mathcal{X}} \frac{p_{\mathcal{A}}\left(I_i | x\right)}{N_i}\right) \leq \frac{\varepsilon}{6}.
$$

For the $T^{th}$ interval,

$$
\sum_{x \in \mathcal{X}} \frac{p_{\mathcal{A}}\left(T | x\right)}{N_T} \leq \frac{k}{N_T} \leq \frac{\varepsilon}{6}.
\tag{39}
$$

Adding the contributions from all the intervals gives us the desired bound. $\qquad\square$

### E.3 Clipping Error Bound

In this part, we will bound the additional bias induced by the clipping step (Step 8 of GENCONDEXP). In particular, we will prove the following lemma:

**Lemma 14.** (*Clipping Error bound*) Let $\hat{H}_{\mathcal{I}}$ be the estimate given by Algorithm 6 and $\hat{H}_{\mathcal{I}}^*$ be the entropy estimate without the clipping step in Algorithm 9, then

$$
\left|\mathbb{E}\left[\hat{H}_{\mathcal{I}}\right] - \mathbb{E}\left[\hat{H}_{\mathcal{I}}^*\right]\right| \leq \frac{\varepsilon}{3}.
\tag{40}
$$

*Proof.* As before, $\bar{H}_i^*$ is the unclipped version of $\bar{H}_i$. Hence we have

$$\left| \mathbb{E}\left[\hat{H}_{\mathcal{I}}^*\right] - \mathbb{E}\left[\hat{H}_{\mathcal{I}}\right]\right| = \left| \sum_{i=1}^{T} p_{\mathcal{A}}(I_i)\mathbb{E}\left[\bar{H}_i\right] - \sum_{i=1}^{T} p_{\mathcal{A}}(I_i)\mathbb{E}\left[\bar{H}_i^*\right]\right|$$

$$= \left| \sum_{i=1}^{T} p_{\mathcal{A}}(I_i)(\mathbb{E}\left[\bar{H}_i\right] - \mathbb{E}\left[\bar{H}_i^*\right])\right| \leq \sum_{i=1}^{T} p_{\mathcal{A}}(I_i)\left|\mathbb{E}\left[\bar{H}_i\right] - \mathbb{E}\left[\bar{H}_i^*\right]\right|. \quad (41)$$

Let's first bound each term $p_{\mathcal{A}}(I_i)\left|\mathbb{E}\left[\bar{H}_i\right] - \mathbb{E}\left[\bar{H}_i^*\right]\right|$ separately. Let $E_{X,i}^* = \log\left(\frac{N_i}{N_{X,i}+1}\right)$ be the unclipped version of $E_{X,i}$ during each round when we are trying to estimate $H_i$. As we can see from the algorithm, $E_{X,i}^*$'s are independent and $\bar{H}_i^*$ is the empirical average of $E_{X,i}^*$'s in the same batch. Hence we know:

$$\mathbb{E}\left[\bar{H}_i^*\right] = \mathbb{E}\left[E_{X,i}^*\right].$$

Similarly,

$$\mathbb{E}\left[\bar{H}_i\right] = \mathbb{E}\left[E_{X,i}\right].$$

Hence we have $p_{\mathcal{A}}(I_i)|\mathbb{E}\left[\bar{H}_i\right] - \mathbb{E}\left[\bar{H}_i^*\right]| = p_{\mathcal{A}}(I_i)|\mathbb{E}\left[E_{X,i}\right] - \mathbb{E}\left[E_{X,i}^*\right]|.$

When $\frac{N_i}{N_{X,i}+1} \geq \frac{1}{4h_i}$, we know $E_{X,i} = \max\{\log\left(\frac{N_i}{N_{X,i}+1}\right), \log\frac{1}{4h_i}\} = E_{X,i}^*$

Next, let's consider the case when $\frac{N_i}{N_{x,i}+1} \in [\frac{N_i}{N_i+1}, \frac{1}{4h_i})$, which is $N_{X,i} \in (4h_iN_i - 1, N_i]$. We divide the interval into $i-1$ intervals, which are

$$L_1 = (4N_ih_2 - 1, N_i], L_2 = (4N_ih_3 - 1, 4N_ih_2 - 1], ..., L_{i-1} = (4N_ih_i - 1, 4N_ih_{i-1} - 1]$$

And the corresponding ranges of $\frac{N_i}{N_{X,i}+1}$ are $[\frac{N_i}{N_i+1}, \frac{1}{4h_2}), [\frac{1}{4h_2}, \frac{1}{4h_3}), ..., [\frac{1}{4h_{i-1}}, \frac{1}{4h_i}).$

Since we are conditioning on $\text{GENESTINT}(\{N_i\}_{i=1}^i, X) = I_i$, $X$ here is distributed according to $p_{\mathcal{A}}(X|I_i)$. Then we can rewrite the difference as:

$$p_{\mathcal{A}}(I_i)|\mathbb{E}\left[E_{X,i}\right] - \mathbb{E}\left[E_{X,i}^*\right]| = p_{\mathcal{A}}(I_i)\mathbb{E}\left[E_{X,i} - E_{X,i}^*\right]$$

$$= p_{\mathcal{A}}(I_i)\sum_{t=1}^{i-1}\sum_{s\in L_t}\sum_{x}\Pr\left(N_{X,i} = s|X = x\right)p_{\mathcal{A}}(x|I_i)(\log\frac{1}{4h_i} - \log\frac{N_i}{s+1})$$

$$\leq \sum_{t=1}^{i-1}p_{\mathcal{A}}(I_i)\sum_{s\in L_t}\sum_{x}\Pr\left(N_{X,i} = s|X = x\right)\frac{p(x)p_{\mathcal{A}}(I_i|x)}{p_{\mathcal{A}}(I_i)}(\log\frac{1}{4h_i} - \log\frac{1}{4h_t})$$

$$\leq \sum_{t=1}^{i-1}p_{\mathcal{A}}(I_i)\sum_{x}\Pr\left(N_{X,i} \in L_t|X = x\right)\frac{p(x)p_{\mathcal{A}}(I_i|x)}{p_{\mathcal{A}}(I_i)}\beta\log^{(t)}(k)$$

$$= \sum_{t=1}^{i-1}\sum_{x}\Pr\left(N_{X,i} \in L_t|X = x\right)p_{\mathcal{A}}(I_i|x)p(x)\beta\log^{(t)}(k). \quad (42)$$

By Chernoff bound, we can get if $p(x) < 2h_{t+1}$,

$$\Pr\left(N_{X,i} \in L_t|X = x\right) \leq \exp\left(-\frac{N_ih_{t+1}}{3}\right).$$

If $p(x) > 2h_{t+1}$,

$$p_{\mathcal{A}}(I_i|x) \leq \exp\left(-\frac{N_i h_{t+1}}{6}\right).$$

Hence

$$\Pr\left(N_{X,i} \in L_t | X = x\right) p_{\mathcal{A}}(I_i|x) \leq \max\{\Pr\left(N_{X,i} \in L_t | X = x\right), p_{\mathcal{A}}(I_i|x)\} \leq \exp\left(-\frac{N_i h_{t+1}}{6}\right)$$

Recall that $N_i = C_N \cdot \frac{k}{\varepsilon(\log^{(i)}(k))^\gamma}, h_i = \frac{(\log^{(i-1)}(k))^\beta}{k}$. Plugging in we get

$$\Pr\left(N_{X,i} \in L_t | X = x\right) p_{\mathcal{A}}(I_i|x) \leq \exp\left(-\frac{C_N \log^{(t)}(k)^\beta}{6\varepsilon \log^{(i)}(k)^\gamma}\right) \leq \frac{6\varepsilon \log^{(i)}(k)^\gamma}{C_N \log^{(t)}(k)^\beta} \leq \frac{6\varepsilon}{C_N \log^{(t)}(k)^\gamma}. \tag{43}$$

Plugging it into Equation 42, we get

$$p_{\mathcal{A}}(I_i)|\mathbb{E}\left[E_{x,i}\right] - \mathbb{E}\left[E_{x,i}^*\right]| \leq \sum_{t=1}^{i-1} \frac{6\varepsilon}{C_N \log^{(t)}(k)^\gamma} \beta \log^{(t)}(k) = \sum_{t=1}^{i-1} \frac{6\beta\varepsilon}{C_N \log^{(t)}(k)^{\gamma-1}}.$$

By (38), we get:

$$p_{\mathcal{A}}(I_i)|\mathbb{E}\left[E_{x,i}\right] - \mathbb{E}\left[E_{x,i}^*\right]| \leq \frac{6\beta\varepsilon}{C_N} \sum_{t=1}^{i-1} e^{t+1-T} \leq \frac{18\beta\varepsilon}{C_N} e^{i-T}$$

Plugging this into (41), and summing over $T$ intervals, we get:

$$\left|\mathbb{E}\left[\hat{H}_{\mathcal{I}}\right] - \mathbb{E}\left[\hat{H}_{\mathcal{I}}^*\right]\right| \leq \frac{18\beta\varepsilon}{C_N} \sum_{i=1}^{T} e^{i-T} \leq \frac{36\beta\varepsilon}{C_N}.$$

Hence we get (40) is true with $C_N > 108\beta$. $\qquad\square$

### E.4 Concentration Bound

In this section, we will derive a high probability bound on $\left|\hat{H}_{\mathcal{I}} - \mathbb{E}\left[\hat{H}_{\mathcal{I}}\right]\right|$. In particular, we will prove the following lemma:

**Lemma 15.** *(**Concentration bound**) Let $\hat{H}_{\mathcal{I}}$ be the estimate given by Algorithm 6 , then*

$$\Pr\left(\left|\hat{H}_{\mathcal{I}} - \mathbb{E}\left[\hat{H}_{\mathcal{I}}\right]\right| > \frac{\varepsilon}{3}\right) \leq \frac{1}{3}. \tag{44}$$

*Proof.*

$$\begin{aligned}
\left|\hat{H}_{\mathcal{I}} - \mathbb{E}\left[\hat{H}_{\mathcal{I}}\right]\right| &= \left|\sum_{i=1}^{T} \hat{p}_{\mathcal{A}}\left(I_i\right) \bar{H}_i - \sum_{t=1}^{T} p_{\mathcal{A}}\left(I_i\right) \mathbb{E}\left[\bar{H}_i\right]\right| \\
&= \left|\sum_{i=1}^{T} \hat{p}_{\mathcal{A}}\left(I_i\right) \bar{H}_i - \sum_{t=1}^{T} p_{\mathcal{A}}\left(I_i\right) \bar{H}_i + \sum_{t=1}^{T} p_{\mathcal{A}}\left(I_i\right) \bar{H}_i - \sum_{t=1}^{T} p_{\mathcal{A}}\left(I_i\right) \mathbb{E}\left[\bar{H}_i\right]\right| \\
&\leq \sum_{i=1}^{T-1} |\hat{p}_{\mathcal{A}}\left(I_i\right) - p_{\mathcal{A}}\left(I_i\right)| \left|\bar{H}_i - \bar{H}_T\right| + \sum_{i=1}^{T} p_{\mathcal{A}}\left(I_i\right) \left|\bar{H}_i - \mathbb{E}\left[\bar{H}_i\right]\right|. \tag{45}
\end{aligned}$$

Next, we will bound each of the term seperately. For the first $T - 1$ terms, note that, because of the clipping step, $\bar{H}_i \in \left[\log\left(\frac{1}{4h_i}\right), \log N_i\right]$. Hence we have:

$$\left|\bar{H}_i - \bar{H}_T\right| \leq \log(4N_T h_i) \leq (\beta + 1) \log \frac{\log^{(i-1)}(k)}{\varepsilon}$$

The estimation of $\hat{p}_{\mathcal{A}}(I_i)$ requires $R_i$ independent executions of GENESTPROBINT. Therefore, by Hoeffding's inequality (Lemma 2), we have

$$\Pr\left(\left|\hat{p}_{\mathcal{A}}(I_i) - p_{\mathcal{A}}(I_i)\right| > t_i\right) \leq 2\exp\left(-R_i t_i^2\right).$$

Choosing $t_i = \dfrac{\varepsilon}{C_T (\log(4N_T h_i))^{5/4}}$ ($C_T \geq 30$ and constant) for $i = 1, \ldots, T-1$ and the value of $R_i$ from Algorithm 6, the right hand expression can be bounded by:

$$2\exp\left(-R_i t_i^2\right) \leq 2\exp\left(-\frac{C_R\left(\log\left(\log^{(i-1)}(k)/\varepsilon\right)\right)^{1/2}}{C_T^2(\beta+1)^{5/2}}\right) \leq \frac{2C_T^4(\beta+1)^5}{C_R^2 \log^{(i)}(k)}. \qquad (46)$$

the last inequality follows from $e^{-x} \leq \frac{1}{x^2}$ for $x > 0$. Combing these for all $T-1$ intervals, let

$$A = \{\sum_{i=1}^{T-1} \left|\hat{p}_{\mathcal{A}}(I_i) - p_{\mathcal{A}}(I_i)\right| \left|\bar{H}_i - \bar{H}_T\right| \geq \sum_{i=1}^{T-1} t_i \left|\bar{H}_i - \bar{H}_T\right|\}.$$

Then by union bound and (38), we have:

$$\Pr(A) \leq \sum_{i=1}^{T-1} \frac{2C_T^4(\beta+1)^5}{C_R^2 \log^{(i)}(k)} \leq \frac{2C_T^4(\beta+1)^5}{C_R^2} \sum_{i=1}^{T-1} \frac{1}{e^{i-1}} \leq \frac{4C_T^4(\beta+1)^5}{C_R^2}.$$

Notice that:

$$\sum_{i=1}^{T-1} t_i \left|\bar{H}_i - \bar{H}_T\right| \leq \sum_{i=1}^{T-1} \frac{\varepsilon}{C_T(\log(4N_T h_i))^{5/4}} \log(4N_T h_i) \leq \sum_{i=1}^{T-1} \frac{\varepsilon}{C_T\left((\beta+1)\log^{(i)}(k)\right)^{1/4}}$$

$$\leq \sum_{i=1}^{T-1} \frac{\varepsilon}{C_T e^{\frac{i-1}{4}}} \leq \frac{5\varepsilon}{C_T} \leq \frac{\varepsilon}{6}.$$

Hence we have for $C_R \geq 6C_T^2(\beta+1)^{5/2}$, we have:

$$\Pr\left(\sum_{i=1}^{T-1} \left|\hat{p}_{\mathcal{A}}(I_i) - p_{\mathcal{A}}(I_i)\right| \left|\bar{H}_i - \bar{H}_T\right| \geq \frac{\varepsilon}{6}\right) \leq \Pr(A) \leq \frac{1}{6}$$

For the second term in Equation (45), we use Lemma 3 where $p = p_{\mathcal{A}}(I_i)$, $m = R_i$, $b = \log N_i$, $a = \log\frac{1}{h_i}$ to get

$$\Pr\left(p_{\mathcal{A}}(I_i)\left|\bar{H}_i - \mathbb{E}\left[\bar{H}_i\right]\right| > c_i\right) \leq 3\exp\left(\frac{-R_i c_i^2}{8p_{\mathcal{A}}(I_i)(\log 4N_i h_i)^2}\right).$$

Take $c_i = \dfrac{\varepsilon}{C_c(\log(4N_T h_i))^{1/4}}$ ($C_T \geq 30$ and constant) for $i = 1, \ldots, T$. Using similar union bound argument as the first part, we get:

$$\Pr\left(\sum_{i=1}^{T} p_{\mathcal{A}}(I_i)\left|\bar{H}_i - \mathbb{E}\left[\bar{H}_i\right]\right| \geq \frac{\varepsilon}{6}\right) \leq \frac{1}{6}.$$

Hence, combining the two, we get:

$$\Pr\left(\left|\hat{H}_{\mathcal{I}} - \mathbb{E}\left[\hat{H}_{\mathcal{I}}\right]\right| > \frac{\varepsilon}{3}\right) \leq \frac{1}{3}.$$

$\square$