[Reviews · NeurIPS 2019]

Reviewer 1



While a huge amount of work exists on sample-complexity of entropy estimation, the space complexity was overlooked. However, it is clear that from the practical point of view, it is important to study the trade-off between sample and space complexities of statistical estimation tasks. So in that sense I consider the main idea of the paper original. Most part of the paper are clear, but I do think that the authors should read their paper carefully and fix some typos/grammar mistakes. The proofs are easy to follow. For me, there is one main disadvantage for this paper: Overall, the trade-off is not clear. The authors just proposed a simple algorithm which has sub-optimal sample complexity and "constant/optimal" memory complexity. However, to really study the complexity a lower bound is needed. It would be much more interesting to come up with a statement that roughly says that: "if one insist on a memory complexity of at most X then sample complexity must be at least Y". Update: reading the detailed author response I now realize that obtaining a lower bound is quite challenging. Thus, I decided to increase my score accordingly.

Reviewer 2



The algorithm is also simple: the basic idea is to choose an element randomly and estimate its frequency by further sampling enough elements. Repeat this process to estimate E[-log X], which is the entropy. A more sophisticated algorithm is built by dividing [0,1] into several subintervals, and estimate the conditional entropy within each subinterval and combine them in the end. The savings come from the intuition that a large frequency can be estimated with fewer samples. Overall, the algorithm is neat and simple, easy to implement. Minor comments: Line 60: “denoted by S(H,k,eps,delta),” -- insert “by” and add a comma at the end of the phrase Line 71: Do not use citations as subjects. Say instead “Paninski showed … [43]”. Such bad style occurs a lot in this section. Line 76: “denoted by H(X^n)” -- insert “by”. Line 107: delete “bit” in “eps = 1 bit” Line 115: At -> In Algorithm 1, line 5: Do you mean S += …? Please edit the typesetting Algorithm 3, line 4: Does the symbol of + above = mean +=? Please use the same notation throughout the paper Algorithm 4, line 4: Is it ESTINT(N_i, x)?

Reviewer 3



Originality: The space-sample tradeoff in memory-constrained statistical inference, especially the lower bounds, is typically very challenging and underexplored until quite recently, e.g., the seminal paper [Raz'16]. The Shannon entropy estimation problem is also a challenging example of high-dimensional functional estimation, which was worked out recently in [Jiao et al.'15, Wu--Yang'16]. This manuscript combines the entropy estimation problem with memory constraints, which I believe to be quite novel. The techniques in the simple algorithm are similar in spirit to [Alon et al.'96], while the authors found a suitable version for entropy estimation. Besides, the authors wrote an excellent literature review in the introduction. Quality: The authors exclusively focused on entropy estimation with constant memory, and worked out an upper bound for the sample complexity. Specifically, a simple algorithm would achieve the sub-optimal bound with an additional log factor, and a careful interval algorithm would fix the log factor. I checked the high-level ideas and most of the proofs, which are all correct. However, I still have some specific comments as follows: 1. It is unfortunate that the authors did not work out a lower bound. Without a matching lower bound, one may doubt whether the considerable amount of effort for the logarithmic improvement in this manuscript is significant or not. I understand that the lower bound may take huge effort and deserves another independent paper, while I feel that the authors should highlight its difficulty. 2. It is also to some extent unsatisfactory that the authors only considered constant memory. From Table I one may be tempted to think that as long as sample complexity * space complexity >= k/eps^3, the entropy estimation problem becomes feasible. Can the proposed idea be generalized to incorporate the case where the space can be larger? 3. The authors first proposed a simple algorithm (Algorithm 1) which has a logarithmic gap from the desired sample complexity. I have the following questions: 3.1. Algorithm 1 relies heavily on the specific form of the Shannon entropy in Eqn. (4). Can the same idea be generalized to other functionals as well, say the power sum function? Besides, in the later interval algorithm the log function also becomes important in the analysis. Will the same idea of local intervals work for other functionals too? 3.2. The logarithmic gap is only an upper bound, and it may happen that this bound is loose. To motivate the necessity of the later interval algorithm, the authors should show that the logarithmic gap is unavoidable in the simple algorithm. I checked simple examples and believe that this lower bound is true. 3.3. One may wonder whether fixing a logarithmic gap would be important, given no good lower bound is known. The authors may at least comment on this issue. 4. I am wondering whether some components in the interval-based algorithm are necessary in theory, or are just artifacts in the proof. Specifically, I am curious about whether the clipping in Algorithms 4 and 9 is indeed necessary. Note that without clipping, the algorithm description could be much clearer: we still use the same estimator in different intervals, and the only difference is that we observe fewer samples for large probabilities, and fewer iterations for small probabilities. I strongly feel that clipping can only be used in the proof, and the same bound holds without clipping in the algorithm. In fact, a simple Chebyshev inequality can be used to bound the concentration term in Eqn. (9), where by triangle inequality the variance can be upper bounded by the sum of the variance of the clipped estimate, and the second moment of the difference. The only remaining problem is about the difference, whose variance seems to be upper bounded by some desirable quantity using the same decomposition arguments as in Appendix F.3. Note that in this way, clipping is only used in the proof but not in the algorithm. 5. Continuing with the previous point, I also feel that some other components of the algorithm may not be really necessary. Note that currently there are lots of wasted/unused samples in Algorithm 4, which seems unnatural. A possibly more natural streaming algorithm would be as follows: after observing some symbol x, first make fewer observations N_1 of the subsequent observations. If the resulting estimate of p(x) is large (i.e., in the largest interval), we simply stop and move on to the next iteration. If it is small, then we make some more observations (N_2 - N_1) to have N_2 observations in total. If the resulting estimate of p(x) is in the second largest interval, we stop; otherwise we continue this process. It seems that the new algorithm shares the same idea of the current one and makes use of samples more efficiently. Will this algorithm work? Clarity: The introduction and the sketch of main ideas are very clear. However, the interval-based algorithm looks a bit messy (due to some theoretical artifacts I think?), which could be hopefully fixed by my above points. Some minor comments: 1. The a += b symbol in the algorithms may be changed into a \leftarrow a + b to avoid ambiguity. 2. In Algorithm 7, line 4-5 seems redundant if we change t-1 to t on line 1. Significance: From the theoretical side, it would be more significant if some lower bound could be proved. However, the algorithms/ideas proposed in this manuscript are quite interesting, and could potentially be generalized to other statistical inference problems with memory constraints. Update: I've read the response. I'm fine with the lower bound issue - as already included in my comments above. However, it's a bit unfortunate to me that the authors did not address the minor technicalities presented above, including the clipping (point 4) and the dependency in the suggested algorithm (point 5). A good paper should be presented in the most natural way and get rid of all unnecessary parts. As a result I will keep my score.

[Author Response · NeurIPS 2019]

We thank all the reviewers for their comments and thoughts on the paper. We particularly thank the reviewers for the encouragement provided about the originality of the work, and for recognizing that this work has the potential of initiating the study of many problems in this field. We also hope that this work will encourage the study of many other statistical problems under limited memory constraints, and hopefully develop tools like Fano's inequality that work under memory constraints. We would like to start by addressing the key concern that most reviewers have, which is not showing a lower bound.

**Comments on the lower bound:** We do believe that the lower bound is not very easy to obtain for this problem, and we have spent time pursuing various approaches to prove a non-trivial lower bound, which were futile. Proving lower bounds even in the streaming model is a hard task, and we believe that the problem is equally hard, if not harder in the statistical setting where we aim to prove trade-off between memory and samples. One of the approaches we pursued was to consider the hard case that was used to prove the sample complexity lower bounds in WuYang, and JiaoHanVenkatWeissman, and reduce the problem to a hypothesis testing problem, which we were unable to prove. We also studied Raz's lower bounds, but currently, do not see a way to apply those techniques for the delicate trade-offs we aim for. In fact, only very recently (COLT 2019, after we submitted the paper), a lower bound on the sample-memory trade-offs were proved in certain regimes, for very specific problem of testing uniformity of distributions. In this light, we expect our work to interest many researchers to study statistical inference tasks under memory constraints, and we hope the reviewers appreciate the same.

We now provide responses to the individual reviewers.

**Reviewer 1** Comment : "Overall, the trade-off is not clear ... must be at least Y"."

Please see the discussion above. We will also go through the paper carefully and fix typos.

**Reviewer 2** We thank the reviewer for their comments and for pointing out typos. We will fix them.

**Reviewer 3** We thank the reviewer for the detailed response and multiple suggestions for improvement.

1) For the lower bound part, please see the discussion above.
2) Extending the algorithm for larger space: We are not sure if there is a trade-off in this case. In fact, one clean open question is to characterize the smallest space needed by an algorithm that is sample optimal, namely uses $k/\varepsilon \log k$ samples. We are unaware of any algorithm that requires a strictly sub-linear space (say $k^{0.1}$ space). Our guess here is that in this case, a sample complexity linear in $k$ is still unavoidable, which can be achieved with constant space using our algorithm. But this again relates to the lower bound part, which we think is hard and leave as future work.
3.1) The ideas in Algorithm 1 as well as the intervals algorithms can indeed be extended to any functional that can be expressed as an expectation of certain functions of the probability mass including power sums. However, we agree the way we select the intervals in this work is more specific to the problem of entropy estimation.
3.2) We agree with the comment, and we did observe that our simple algorithm does need the $\log^2(k)$ term. Consider a distribution that has one element with probability $1/2$, and the remaining elements have probability about $1/2k$. This will correspond to estimating the mean of a distribution that takes values $\log(2)$, and $\log(2k)$ with equal probability, thus requiring $\log^2(k)$ iterations.
3.3) This is a good question, and again relates to the lower bound remark discussed above. In addition, we think that if it is possible to solve general problems with much smaller space, and a tiny overhead in samples, they might be worth considering.
4) This is a great point. We thank the reviewer for pointing this out. We use the clipping step so that the range of our final estimate can be bounded better with fewer iterations. If we use the unclipped estimate as our final estimate, we would have to bound the concentration of the unclipped estimate with more interations. For example, in the two-intervals algorithm, because of the clipping step the range is reduced to $\log(N_i) - \log(1/4l_i) = \log(4N_i l_i)$. The range of the unclipped estimate can only be bounded by $\log(N_i) - \log(N_i/(N_i + 1)) = \log(N_i + 1)$. This might be a trick that is only useful in the proof but currently we don't see an easy way to bound the concentration of the unclipped estimate without using super-linear number of samples. We will further investigate this line of thought.
5) The suggested algorithm would still work, but analyzing the performance would require more considerations because estimates from one interval are now dependent on the estimates from the other intervals. We agree that our algorithms are not sample efficient in terms of constants. However we would like to emphasize that our algorithms have the same sample complexity as that of the improvements suggested.

We thank the reviewers again. Hopefully we have answered their questions and concerns. We hope that the new line of questions proposed in this work takes precendence in the decision over the shortcomings.

[Meta-Review · NeurIPS 2019]

The reviewers read the authors feedback and realized that the lower bound was actually challenging to obtain. They decided that this paper was worth accepting as it contains a significant amount of novelty in the proposed lower bound. Note that it was noted that the authors didn't fully validate some of the concerns, and it would be useful to take them into consideration in the final camera-ready version of the paper: "For streaming algorithms, time lower bounds are pretty hard and this paper achieve a constant space upper bound (and so it can be said to be tight up to a constant factor in terms of space complexity). How serious is the lower bound thing?" "What is the significance of the logarithmic improvement, where people may just like a suboptimal but simple algorithm which is only off a log factor of another possibly suboptimal but complicated algorithm?"